# Current Research on the Pathogenesis of NAFLD/NASH and the Gut–Liver Axis: Gut Microbiota, Dysbiosis, and Leaky-Gut Syndrome

**DOI:** 10.3390/ijms231911689

**Published:** 2022-10-02

**Authors:** Takashi Kobayashi, Michihiro Iwaki, Atsushi Nakajima, Asako Nogami, Masato Yoneda

**Affiliations:** Department of Gastroenterology and Hepatology, Yokohama City University Graduate School of Medicine, 3-9 Fukuura, Kanazawa-ku, Yokohama 236-0004, Japan

**Keywords:** non-alcoholic fatty liver disease, gut–liver axis, gut microbiota, dysbiosis, leaky-gut syndrome

## Abstract

Global lifestyle changes have led to an increased incidence of non-alcoholic fatty liver disease (NAFLD) and non-alcoholic steatohepatitis (NASH), requiring further in-depth research to understand the mechanisms and develop new therapeutic strategies. In particular, high-fat and high-fructose diets have been shown to increase intestinal permeability, which can expose the liver to endotoxins. Indeed, accumulating evidence points to a link between these liver diseases and the intestinal axis, including dysbiosis of the gut microbiome and leaky-gut syndrome. Here, we review the mechanisms contributing to these links between the liver and small intestine in the pathogenesis of NAFLD/NASH, focusing on the roles of intestinal microbiota and their metabolites to influence enzymes essential for proper liver metabolism and function. Advances in next-generation sequencing technology have facilitated analyses of the metagenome, providing new insights into the roles of the intestinal microbiota and their functions in physiological and pathological mechanisms. This review summarizes recent research linking the gut microbiome to liver diseases, offering new research directions to elucidate the detailed mechanisms and novel targets for treatment and prevention.

## 1. Introduction

The liver is directly connected to the intestinal tract through the portal vein and cooperates with the digestive tract to assist with digestion and absorption. The liver plays an important role in the body’s defense system and in immune and allergic mechanisms [1]. The small intestine and liver cross-talk in a multifactorial manner and are involved in the onset and exacerbation of various diseases [2,3]. The Russian Nobel Prize-winning scientist Mechnikov advocated that yogurt (which contains lactobacilli) is useful for longevity, which led to the concept of probiotics being proposed at the beginning of this century.

Although intestinal bacteria have attracted increasing attention, most of them are anaerobic, which hinders their cultivation, identification, and analysis [4]. In recent years, the establishment of new molecular biology methods, especially next-generation sequencing, has facilitated simple analyses.

The prevalence of non-alcoholic fatty liver disease (NAFLD), a major public health concern, is increasing worldwide. However, the pathogenesis of the disease remains unclear, and no therapeutic agents have been developed [5]. The gut–liver axis is involved in NAFLD pathogenesis through various pathways and is a potential novel therapeutic target [6,7]. In this article, we review the existing literature on NAFLD, dysbiosis, leaky-gut syndrome, probiotics, and other potential therapies for the intestinal axis.

## 2. NAFLD/NASH and the Involvement of Intestinal Bacteria in Their Pathogenesis

Viral hepatitis, such as hepatitis B and hepatitis C, used to be the most common cause of chronic hepatitis and cirrhosis, but its prevalence is decreasing with the establishment of diagnostic methods and therapeutic agents. NAFLD and non-alcoholic steatohepatitis (NASH) are the most common liver diseases, with the number of patients increasing rapidly worldwide because of the increasing prevalence of obesity and type 2 diabetes [8,9]. NAFLD is classified into non-alcoholic fatty liver (NAFL), which has a relatively good prognosis with a slow progression of fibrosis, and NASH, which is associated with hepatocellular damage and liver fibrosis [8,9]. According to a meta-analysis, the prevalence of NAFLD is 27.4% in Asia [10] and 22.3% in Japan [11], of which 10–20% is considered NASH.

Dysbiosis and its metabolite, short-chain fatty acids, has long been assumed to play an important role in the development and progression of NAFLD/NASH [12,13]. However, the evidence is not sufficient, and further investigation is expected in this area. Numerous pathways are involved in the pathogenesis of NASH, including lipotoxicity, insulin resistance, oxidative stress, and mitochondrial injury, hindering the development of therapeutic agents owing to the lack of targeted therapy [14].

The primary treatment for NASH is weight loss through lifestyle modification and no specific therapeutic agents are currently available [15]. Recently, intestinal permeability and gut-derived substances have attracted increasing attention as targets for treating NASH.

## 3. Abnormalities of the Intestinal Microbiota and NAFLD/NASH

Several reports have shown the relationship between NAFLD and the qualitative abnormalities of intestinal bacteria [16,17,18,19,20]. However, bacterial flora analyses have not shown consistent results owing to the influence of genetic and environmental factors, such as race and diet, respectively. Further, all reports were based on a small number of patients, and the relationship of bacterial flora with clinical data remains unclear.

Miele et al. [21] studied 35 patients with NAFLD and found increased intestinal permeability and small intestinal bacterial overgrowth (SIBO) in the NAFLD group (60% in the NAFLD group and 20.8% in the control group; *p* < 0.001). Intestinal permeability and SIBO were reported to be correlated with the severity of hepatic steatosis. SIBO is thought to cause liver inflammation via endotoxemia [22,23]. Lichtman et al. [24] reported that antibiotics ameliorated liver damage in rats with surgically induced intestinal bacterial overgrowth. Bergheim et al. [25] also reported that antibiotics ameliorate liver steatosis and endotoxemia in a rodent NAFLD model induced by fructose. As discussed in the next section, NAFLD is known to be exacerbated when the liver is exposed to endotoxins. These findings suggest an important role for the small intestinal microbiota in NAFLD and suggest that treating SIBO may improve endotoxemia and NAFLD. 

Moreover, alterations in the gut microbiota inhibit the synthesis of fasting-induced adipocyte factor (FIAF), also known as angiopoietin-related protein 4. FIAF specifically inhibits lipoprotein lipase, an enzyme involved in the release of free fatty acid from very low-density lipoprotein particles to the liver. Thus, changes in the intestinal microbiota have been associated with fat accumulation in the liver [26,27].

## 4. Role of Endotoxin in NAFLD/NASH Pathogenesis

NASH is characterized by the accumulation of hepatic fat and chronic inflammation with neutrophil infiltration. Changes in the intestinal environment are attracting attention as a mechanism for the induction of chronic inflammation. Intestinal environment deterioration because of diet westernization leads to an increase in anaerobic Gram-negative rods and an increase in blood levels of lipopolysaccharide (LPS), a component of the outer wall membrane of Gram-negative rods and an endotoxin (metabolic endotoxemia) [28]. In patients with NAFLD/NASH who consume high-fat and high-fructose diets, intestinal permeability is increased, as described in the next section, inducing leaky-gut syndrome; therefore, endotoxin is assumed to be exposed to the liver via the portal vein [27].

Patients with NASH have higher levels of endotoxin in their blood than healthy individuals [29]. Endotoxin reacts with a group of receptors such as toll-like receptors (TLRs) and nucleotide-binding oligomerization domain receptors. In particular, TLR4 is expressed on the plasma membrane of hepatocytes and Kupffer cells, and TLR4-mediated signals are thought to activate signaling molecules such as nuclear factor kappa B, which induces the production of inflammatory cytokines (interleukin [IL]-1β and IL-18) and liver injury. The intestinal environment is thought to play an important role in the pathogenesis and progression of NAFLD/NASH, including increased intestinal permeability, qualitative and quantitative abnormalities of intestinal bacteria, and increased endogenous alcohol production [6,27].

## 5. Increased Endogenous Alcohol Production by Intestinal Bacteria and NAFLD/NASH Pathophysiology

Zhu et al. [16] investigated the differences in gut microbiota and blood ethanol concentrations between patients who were obese but without NASH (non-NASH) and those with NASH and obesity. The authors reported that patients with NASH and obesity had higher blood ethanol concentrations than those in the non-NASH group. Furthermore, ethanol-producing bacteria, such as Enterobacteriaceae and *Escherichia* (phylum Proteobacteria), were increased in patients with NASH and obesity. Thus, endogenous alcohol is suggested to contribute to NASH progression. Endogenous alcohol is not only directly toxic to the liver but overproduction of alcohol activates the ethanol metabolic pathway in the liver, increasing oxidative stress in the liver [30].

## 6. Hepatocyte Endotoxin Hypersensitivity in NAFLD/NASH Development

As mentioned above, patients with NAFLD and NASH are likely to have an environment in which factors from the intestinal side, such as qualitative and quantitative abnormalities of intestinal bacteria, increased intestinal permeability, or increased endogenous alcohol, are likely. However, it is difficult to explain the factors that cause NASH and fibrosis to develop. In fact, low levels of endotoxin are present in the portal vein even in healthy subjects [31]. Therefore, it is speculated that endotoxin responsiveness is different between hosts, i.e., in the liver of patients with NAFLD/NASH, endotoxin is an important attack factor in NAFLD/NASH progression.

In a mouse model of obesity, hyperleptinemia associated with obesity is reported to activate STAT3 signaling in Kupffer cells, which in turn increases the expression of CD14, a co-receptor of TLR4, and induces an excessive inflammatory response of the liver to endotoxin, leading to the development of NASH pathology [32]. These findings suggest that obesity and a high-fat diet are deeply involved in NALFD/NASH pathogenesis from both the hepatic and intestinal routes, as they increase the amount of LPS influx into the liver via SIBO and leaky-gut syndrome in the small intestine, along with the formation of intrahepatic lipidation in NAFLD/NASH. They are also involved in the excessive response to LPS in the liver.

## 7. NAFLD/NASH and Intestinal Metabolites

Boursier et al. [20] classified NAFLD into fibrosis stages 0–1 and 2–4 and performed predictive metagenomic analysis against the Kyoto Encyclopedia of Genes and Genomes (KEGG) pathway. Carbohydrate metabolism included glyoxylate and dicarboxylate metabolism, pentose and glucuronate interconversions, and the pentose phosphate pathway, while lipid metabolism included fatty acid biosynthesis and lipid biosynthesis proteins. Based on imputed metagenomic profiles, the Kyoto Encyclopedia of Genes and Genomes pathways were significantly related to NASH and fibrosis F ≥ 2 were mostly related to carbohydrate, lipid, and amino acid metabolism. They identified Bacteroides as independently associated with NASH and *Ruminococcus* with significant fibrosis. Loomba et al. [19] performed metagenomic analyses (functional analysis based on the genetic information of intestinal bacteria) using whole genome sequencing and showed that enzymes related to the production of short-chain fatty acids such as butyrate, D-lactate, propionate, and acetate were abundant in patients with advanced fibrosis. Further, Loomba et al. [19] reported that alcohol dehydrogenase, an enzyme related to ethanol metabolism, was increased in patients with advanced fibrosis, whereas alcohol dehydrogenase NADP(+) was increased in patients with mild fibrosis, suggesting a controversial effect of endogenous alcohol. Further analyses, such as fecal metabolome analysis, are needed to gain further insights.

Bile acids, an important intestinal metabolite, are broadly classified into primary bile acids, which are synthesized directly from cholesterol in the liver and secondary bile acids, which are dehydroxylated by intestinal bacteria after the primary bile acids are excreted into the intestinal tract. The primary bile acids include cholic acid (CA) and chenodeoxycholic acid (CDCA), whereas the dehydroxylated secondary bile acids include deoxycholic acid (DCA) and lithocholic acid (LCA). These bile acids are conjugated with taurine or glycine by bile acid CoA:amino acid N-acyltransferase and secreted into bile as conjugated bile acids. LCA, DCA, and CDCA are highly hydrophobic and cytotoxic among bile acids [33]. In contrast, ursodeoxycholic acid is a hydrophilic bile acid and is known as a cellular bile acid with almost no cytotoxicity.

Bile acids assist in the digestion and absorption of lipids and fat-soluble vitamins. In ob/ob mice, a leptin-deficient obese insulin-resistant type 2 diabetes model, leptin administration increases hydrophilic bile acids and decreases the bile acid pool [34]. Although the synthesis and secretion of bile acids, especially CA, DCA, CDCA, and LCA, is reported to be enhanced in Native American type 2 diabetic Pima Indians, their secretion is decreased by insulin administration. However, the role of bile acids in the pathogenesis of insulin-resistant type 2 diabetes remains poorly understood [35].

Bile acids also prevent intestinal bacterial overgrowth and exert a strong antimicrobial effect on intestinal homeostasis [36]. This protective effect is provided by the detergent action and farnesoid X receptor (FXR) activation. Regarding the pathogenesis of NAFLD, bile acids have been shown to regulate lipid metabolism by binding to FXR [37]. Bile acid-mediated FXR activation in the liver was reported to induce the expression of the atypical nuclear receptor, a small heterodimer partner, which promotes inhibition of SREBP-1c, thereby suppressing triglyceride synthesis in the liver [38].

Recently, Kasai et al. [39] simultaneously evaluated both blood and fecal bile acids in patients with NAFLD. In the NAFLD group with worsening fibrosis, bile acid concentrations in the feces and serum and the level of 7α-hydroxy-4-cholesten-3-one—a surrogate marker for bile acids synthesis—were higher than NAFLD with mild fibrosis, suggesting the involvement of bile acids. This suggests that abnormal bile acids metabolism may be a potential therapeutic target for NAFLD with fibrosis.

## 8. Peroxisome Proliferator-Activated Receptors (PPARs) and the Gut Microbiota in NAFLD

PPARs, members of the nuclear receptor superfamily, play an important role in the pathogenesis of NAFLD. Three isoforms of PPARs are known, including PPARα, PPARβ/δ, and PPARγ. Among them, PPARα and PPARγ are notable in NASH. Various lipid metabolic processes, including fatty acid uptake, β-oxidation, and ketogenesis, are regulated by PPARα [40]. In NAFLD, activation of PPARα was reported to improve hepatic pathological findings, including hepatic lipidification [41,42]. PPARγ is involved in lipid metabolism, insulin resistance, and regulation of immune-inflammatory responses. PPARγ expression also improved hepatic steatosis, inflammation, and fibrosis in NAFLD [43,44,45,46,47]

Furthermore, the gut microbiota was reported to affect PPARs in different organs, including the liver [48]. Ruan et al. [49] analyzed the gut microbiota of Dusp6-deficient mice, which are resistant to diet-induced obesity, and found that these mice possess unique gut microbiota. By performing gut transcriptome analysis, they observed that this gut microbiota induces the PPARγ pathway, increased energy expenditure, and reduced body weight gain. In addition, administration of *Lactobacillus casei* in mice enhanced PPARʊ activity and suppressed TLR-4 signaling in the liver, resulting in the suppression of hepatic steatosis [50]. Crawford et al. [51] showed that PPARɑ is involved in the effect of gut microbiota on post-fasting ketosis. Furthermore, Chiu et al. [52] observed that compared to mice transplanted with stools from healthy individuals, liver steatosis and inflammation were exacerbated, and PPARʊ expression was altered in mice transplanted with stools from patients with NASH. These reports suggest that the composition of the gut microbiota influences the pathogenesis of NAFLD, which may be related to the PPAR pathway.

## 9. NAFLD/NASH and Intestinal Permeability

The intestinal mucosa has a very large area spanning approximately 400 m^2^ and is responsible for most of the functions of the small intestine [53]. The mucosal epithelium takes in nutrients, water, and electrolytes from consumed food and simultaneously acts as a barrier to prevent harmful substances such as pathogens, viruses, and up to 100 trillion intestinal bacteria from entering the intestinal tract. The key to this barrier function, which plays a major role in the complex selection process, is the formation of tight junctions and the mucus layer, which are formed by various intercellular adhesion molecules. The composition of tight junctions controls the substances that pass through the epithelial cells, and this composition changes in response to various conditions, including changes in the intestinal microbiota [39,54].

When the intestinal barrier function fails and intestinal permeability is excessively increased, the body is exposed to substances that should not pass through the intestinal tract into the body. This condition is called leaky-gut syndrome. When a food antigen of a size that would not normally enter the body enters the body because of leaky-gut syndrome, it is recognized as a foreign substance and antibodies are produced by the immune response system, resulting in an acquired food allergy. The production of antibodies to large molecular weight food antigens, microorganisms, and metabolites is also thought to cause allergic symptoms to these substances, as well as autoimmune diseases. Therefore, leaky-gut syndrome has attracted attention for inflammatory bowel disease, irritable bowel syndrome, and other intestinal pathologies, as well as its association with food allergies, collagen diseases, diabetes, atherosclerosis, liver diseases, and other diseases. Several factors cause leaky-gut syndrome in our daily life, such as obesity, a high-fat diet, alcohol consumption, fructose intake, SIBO, qualitative and quantitative changes in intestinal bacteria (antibiotics), and medications (aspirin, NSAIDs, and PPIs). Measurement of the urinary lactulose/mannitol ratio is currently the most widely used method for diagnosing leaky-gut syndrome [55].

A significantly higher rate of SIBO was reported to be diagnosed in obese subjects with BMI > 40 (17%) compared to healthy subjects (2.5%) [56]. A study using human-derived Caco-2 cells verified that LPS suppresses the expression of tight junction molecules [57] and that a high-fat diet and bile, both of which cause obesity, similarly suppress the expression of tight junction molecules [58], which may be part of the mechanism linking obesity and leaky-gut syndrome. Recently, the percentage of *Akkermansia muciniphila*, belonging to the phylum Verrucomicrobia, was reported to be decreased in obese and diabetic patients and their mouse models; further, *A. muciniphila*, whether viable or dead, was associated with enhancement of the intestinal mucosal barrier, increased goblet cells, and improved metabolic functions [21].

Patients with NAFLD/NASH often have leaky-gut syndrome. Recently, a meta-analysis confirmed that liver damage precedes increased intestinal permeability, suggesting that patients with NASH tend to have higher intestinal permeability than healthy subjects and that the “gut-liver axis,” wherein intestinal bacteria and bacterial products transferred to the bloodstream aggravate the liver, may be a factor in the development of NASH [28].

## 10. Using the Characteristics of the Gut Microbiota to Diagnose NAFLD-Derived Cirrhosis and Hepatocellular Carcinoma

Recently, intestinal bacteria have been investigated as a diagnostic tool for NAFLD/NASH. In 2019, Caussy et al. [59] performed 16S rRNA metagenomic analysis in 203 patients with NAFLD and found that patients who progressed to cirrhosis presented a characteristic gut microbiota. In this study, a combined panel of multiple species of intestinal bacteria allowed differentiation between NAFLD-derived cirrhosis and non-cirrhosis with a very good diagnostic performance (AUROC: 0.92). Liver biopsy, which is considered the gold standard for diagnosing cirrhosis, has problems such as invasiveness and sampling errors; therefore, gut microbiota analysis is expected to be developed as a new testing tool for cirrhosis [60,61].

In 2019, Ponziani et al. [62] compared the gut microbiota among 21 cases of NAFLD-derived cirrhosis with hepatocarcinoma, 20 cases of NAFLD-derived cirrhosis without hepatocarcinoma, and 20 healthy subjects using 16S rRNA metagenomic analysis. The results showed that Enterobacteriaceae and *Streptococcus* were abundant and that *Akkermansia* were decreased in all patients with cirrhosis. Additionally, Bacteroides and Ruminococcaceae were increased and *Bifidobacterium* was decreased in the hepatocarcinoma group. A characteristic gut microbiota profile was reported in NAFLD-derived cirrhosis and in patients with hepatocellular carcinoma. Further, Ren et al. [63] analyzed the gut microbiota of 419 patients and reported fewer butyrate-producing bacteria and more LPS-producing bacteria in patients with early-stage liver cancer compared to those in non-cancer patients. These data indicated that a panel of 30 combined intestinal bacteria could diagnose the presence or absence of early-stage hepatocellular carcinoma with high accuracy (AUROC 0.806). This suggests that intestinal bacteria may be involved in the development of hepatocellular carcinoma and that diagnosis of early-stage cancer may be facilitated by the analysis of intestinal bacteria.

## 11. NAFLD/NASH Treatment Associated with Intestinal Permeability and Gut Microbiota 

The therapeutic effect of probiotics on NAFLD has been reported previously. Loguercio et al. [64,65] reported that probiotic administration reduced liver injury in patients with NAFLD. Further, a double-blind randomized controlled trial examining the effect of *Lactobacillus bulgaricus* and *Streptococcus thermophiles* in 30 adults with NAFLD and in the placebo control reported that administering *Lactobacillus bulgaricus* and *Streptococcus thermophiles* decreased the transaminase levels [66]. Further, in a double-blind randomized controlled trial of pediatric obese patients with NAFLD, and had an average age of 10.7 years, administration of *Lactobacillus rhamnosus* strain GG for 8 weeks was reported to improve aspartate aminotransferase levels [67].

Kessoku et al. [68] reported a parallel, three-arm, double-blind, randomized trial in 2020 using lubiprostone. Among 150 Japanese patients with NAFLD and constipation, the lubiprostone group showed improvement in the urinary lactulose/mannitol ratio, which is an index of intestinal permeability, along with significant improvement in liver enzymes, hepatic steatosis measured using magnetic resonance imaging-proton density fat fraction and blood endotoxin concentrations. In particular, marked reductions in transaminases, liver fat contents, and blood endotoxin concentrations were observed in the lubiprostone group with improved intestinal permeability.

Further, in 2019, Duseja et al. [69] reported that treatment with a multistrain probiotic (a combination of many bacterial species, including lactobacilli and bifidobacteria) for one year significantly reduced ballooning degeneration and fibrosis in liver biopsies compared to those in a placebo group. Blood endotoxin, inflammatory cytokines, ALT, and other markers were also reported to be significantly decreased. Thus, novel therapeutic agents targeting intestinal permeability and intestinal bacteria in NAFLD/NASH are attracting increasing attention.

## 12. Cirrhosis, Hepatic Encephalopathy, and Intestinal Bacteria

The intestinal barrier function is impaired in patients with cirrhosis, and bacterial translocation is likely to occur. Further, patients with liver cirrhosis are susceptible to infection owing to decreased immune function; thus, appropriate infection control is essential for improving prognosis. Treatment of hepatic encephalopathy is one of the key unmet needs in cirrhosis care. Intestinal bacteria are closely related to the pathogenesis of hepatic encephalopathy, and several therapies for hepatic encephalopathy targeting intestinal bacteria have been reported [70].

The prebiotic lactulose promotes the growth of lactobacilli and bifidobacteria, which are considered beneficial bacteria. Fermentation of lactulose by these bacteria requires amino acid synthesis using ammonia as a substrate, resulting in lower ammonia levels in the intestinal tract. Dhiman et al. [71] performed a meta-analysis of 25 randomized controlled trials and compared lactulose, rifaximin, probiotics, and L-ornithine L-aspartate as treatments for subclinical hepatic encephalopathy. They found that only lactulose was associated with “recovery of subclinical hepatic encephalopathy” and the only drug that met all three endpoints of “recovery from latent hepatic encephalopathy,” “prevention of manifestation of hepatic encephalopathy,” and “improvement in the quality of life.

With regard to probiotics, improvement of intestinal barrier function was considered a key mechanism for improving hepatic encephalopathy. However, in the aforementioned meta-analysis, probiotics were less effective than lactulose in all endpoints, even though they reduced the blood ammonia levels [71]. Insufficient evidence for the use of probiotics against hepatic encephalopathy is available at present and further research is warranted in this area. However, *Escherichia coli* SYNB1020, which was genetically engineered to convert ammonia to L-arginine, was shown to successfully reduce blood ammonia levels in preclinical studies and was also well tolerated in phase I studies [72].

Fecal microbiota transplantation (FMT) is the transplantation of a healthy donor’s feces encapsulated in a capsule to a recipient. FMT has recently been actively investigated for treating hepatic encephalopathy. Bajaj et al. [73] reported a phase I study comparing FMT capsules derived from healthy donors with a placebo. In this study, oral administration of FMT capsules was found to be safe and well tolerated in patients with hepatic encephalopathy and was found to improve performance in cognitive function tests (EncephalApp). However, the performance in another cognitive function test (psychometric HE score) did not improve and no significant difference in the rate of hepatic encephalopathy manifestation at 5 months of follow-up was observed. Therefore, the authors concluded that further studies are needed to prove the efficacy of FMT. Several other studies evaluating FMT for hepatic encephalopathy have been reported, but all of them are small and in Phase I trials. Further, FMT involves many issues, including infectivity, the uncertainty of efficacy, selection of appropriate donors, and stability of distribution, that should be resolved before its clinical application.

Antibiotics are used for treating hepatic encephalopathy as a means of reducing ammonia-producing bacteria. Currently, no antibiotic has demonstrated efficacy and safety comparable to that of rifaximin, making rifaximin the first choice for treatment. In 2019, Salehi et al. [74] performed a retrospective study on patients on the liver transplant waiting list. Results indicated that patients who were prescribed rifaximin for hepatic encephalopathy had lower hospitalization rates, SBP incidence, and gastroesophageal varix rupture rates compared to those in patients who were not prescribed the drug. Further, rifaximin-treated patients reported shorter hospital stays and longer time to rehospitalization.

Bacteriophages are being studied as a new means of reducing harmful intestinal bacteria. Bacteriophages are viruses that specifically attack bacteria and have been reported to eliminate cytolysin-producing *Enterococcus faecalis*, a major pathogen in alcoholic hepatitis [75]. Bajaj et al. [73] also reported that the amount of bacteriophages in stool correlated with the MELD score and severity of hepatic encephalopathy in cirrhotic patients. Therefore, treatment of hepatic encephalopathy with bacteriophages shows promise.

## 13. Conclusions

Dysbiosis of intestinal microbiota, such as increased intestinal permeability, qualitative and quantitative abnormalities of intestinal bacteria, and increased endogenous alcohol consumption, as well as increased sensitivity to endotoxin as a factor of the host liver, are thought to be involved in the pathogenesis of NAFLD and NASH (Figure 1). The mechanism of pathogenesis is considered to involve both liver and intestinal factors. Further, abnormalities in intestinal metabolites suggest differences in enzymes related to short-chain fatty acids, carbohydrate metabolism, and lipid metabolism. Further studies are needed to elucidate the pathogenesis of NAFLD from new perspectives focusing on abnormalities of the intestinal microbiota and metabolites and to develop treatment methods based on these findings.

## Figures and Tables

**Figure 1 ijms-23-11689-f001:**
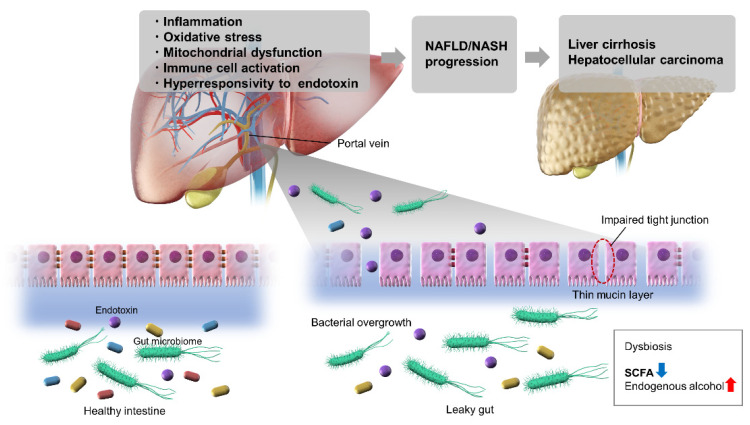
Schematic diagram of the gut–liver axis in non-alcoholic fatty liver disease/non-alcohol fatty liver disease. NAFLD, non-alcoholic fatty liver disease; NASH, non-alcoholic steatohepatitis; SCFA, short chain fatty acids.

## Data Availability

Not applicable.

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
