# Peer review of "Current Research on the Pathogenesis of NAFLD/NASH and the Gut–Liver Axis: Gut Microbiota, Dysbiosis, and Leaky-Gut Syndrome"

_ijms, 2022, doi:10.3390/ijms231911689_

Round 1
Reviewer 1 Report
This is my opinion on “Current Research on the Pathogenesis of NAFLD/NASH and Gut-liver Axis: Gut microbiota, Dysbiosis, and Leaky-Gut Syndrome”. This review assesses recent research regarding the correlation between the gut microbiome and liver diseases.
Introduction need revisions and it lacks of appropriate citations. You should add references and highlight the reasons for conducting this review.
Lines 57-58 need to be cited.
Lines 79-82 and 86-92 need to be cited.
Lines 153-160 need to be cited or excluded as it does not offer significant novel knowledge.
Lines 230-233 need extra references regarding the gold-standard and its cons, especially for pediatric patients. Authors should add this reference https://pubmed.ncbi.nlm.nih.gov/34201230/
Lines 278-284 need to be cited.
This is an interesting review. However, authors presented the manuscript as a book chapter than a scientific paper. More references need to be added in order to strengthen their findings and add a scientific soundness.
Author Response
Dear Prof. Dr. Maurizio Battino, Editor-in-Chief, International Journal of Molecular Sciences
September 15th, 2022
We would like to express our sincere gratitude to both you and the Reviewers for taking the time to review our manuscript titled “Current Research on the Pathogenesis of NAFLD/NASH and Gut-liver Axis: Gut microbiota, Dysbiosis, and Leaky-Gut Syndrome.” submitted to International Journal of Molecular Sciences. The reviewers’ comments and Editor’s comments were very encouraging and helpful. We have now amended our manuscript according to your suggestions and the Reviewer’s comments. Please find below the changes we have made to the manuscript and our answers to the reviewers’ comments in a point-by-point manner.
Response to Reviewer 1
Thank you very much for your useful suggestions. Your suggestions have been fully addressed in the revised manuscript, which we feel has been now greatly improved as a result.
Comment 1.
Introduction need revisions and it lacks of appropriate citations. You should add references and highlight the reasons for conducting this review.
Response to comment 1.
I appreciate your advice, we have revised the introduction and added citations.
Comment 2.
Lines 57-58 need to be cited.
Response to comment 2.
Thank you for the advice, we have added reference.
comment 3.
Lines 79-82 and 86-92 need to be cited.
Response to comment 3.
Thank you for the advice, we have added reference.
Comment 4.
Lines 153-160 need to be cited or excluded as it does not offer significant novel knowledge.
Response to comment 4.
I appreciate your advice, we have removed the corresponding section.
Comment 5.
Lines 230-233 need extra references regarding the gold-standard and its cons, especially for pediatric patients. Authors should add this reference https://pubmed.ncbi.nlm.nih.gov/34201230/
Response to comment 5.
Thank you for the advice, we have added the references you suggested.
Comment 6.
Lines 278-284 need to be cited.
Response to comment 6.
I appreciate your advice, we have added citations.

Reviewer 2 Report
This review article reviews the relationship between NAFLD/NASH and the small intestinal environment and microbiome, and I believe that considering the development of NAFLD/NASH from the small intestinal environment is an important perspective regarding the prevention of NAFLD/NASH.
Throughout, it is not clear how each of the phenomena discussed in chapters 2 through 11 are linked to the development of NAFLD/NASH. This is because it is not clear how each factors in small intestine is related to lipid synthesis pathways, inflammation, and oxidative stress in the liver in the pathogenesis of NAFLD/NASH. Please also mention the molecular mechanisms in these pathways in the liver. For example, why small intestinal permeability and SIBO cause hepatic steatosis in chapter 3, and what mechanisms exist that make endotoxin bad in chapter 4. Please describe what exactly happens in the liver as a result of the factors discussed in chapters 2 through 11 in the small intestine. This would make the review interesting to readers studying NAFLD/NASH.
Give a reference to the area of the small intestinal mucosa in the first sentence of chapter 8.
Author Response
Dear Prof. Dr. Maurizio Battino, Editor-in-Chief, International Journal of Molecular Sciences
September 15th, 2022
We would like to express our sincere gratitude to both you and the Reviewers for taking the time to review our manuscript titled “Current Research on the Pathogenesis of NAFLD/NASH and Gut-liver Axis: Gut microbiota, Dysbiosis, and Leaky-Gut Syndrome.” submitted to International Journal of Molecular Sciences. The reviewers’ comments and Editor’s comments were very encouraging and helpful. We have now amended our manuscript according to your suggestions and the Reviewer’s comments. Please find below the changes we have made to the manuscript and our answers to the reviewers’ comments in a point-by-point manner.
Response to Reviwer 2.
Comment 1.
This review article reviews the relationship between NAFLD/NASH and the small intestinal environment and microbiome, and I believe that considering the development of NAFLD/NASH from the small intestinal environment is an important perspective regarding the prevention of NAFLD/NASH.
Throughout, it is not clear how each of the phenomena discussed in chapters 2 through 11 are linked to the development of NAFLD/NASH. This is because it is not clear how each factors in small intestine is related to lipid synthesis pathways, inflammation, and oxidative stress in the liver in the pathogenesis of NAFLD/NASH. Please also mention the molecular mechanisms in these pathways in the liver. For example, why small intestinal permeability and SIBO cause hepatic steatosis in chapter 3, and what mechanisms exist that make endotoxin bad in chapter 4. Please describe what exactly happens in the liver as a result of the factors discussed in chapters 2 through 11 in the small intestine. This would make the review interesting to readers studying NAFLD/NASH.
Response to comment 1.
I appreciate your advice. In Chapter 3, we added the mechanism by which SIBO causes liver steatosis. In Chapter 4, we have described the molecular mechanisms involved, including the involvement of TLR4. We have also added information on the respective mechanisms in Chapters 5 and 8.
Comment 2.
Give a reference to the area of the small intestinal mucosa in the first sentence of chapter 8.
Response to comment 2.
Thank you for the advice, we have added a citation.

Round 2
Reviewer 2 Report
Thanks to the addition of the molecular mechanism, I think it is easier to understand the pathogenesis of NAFLD. Not only SREBP-1c but also PPARgamma is involved in the pathogenesis of NAFLD. PPARalpha is also involved in preventing NAFLD development. Showing the relationship between these molecules in the liver and the small intestinal environment will make a more complete paper.
Author Response
Dear Prof. Dr. Maurizio Battino, Editor-in-Chief, International Journal of Molecular Sciences
September 27th, 2022
We would like to express our sincere gratitude to both you and the Reviewers for taking the time to review our manuscript titled “Current Research on the Pathogenesis of NAFLD/NASH and Gut-liver Axis: Gut microbiota, Dysbiosis, and Leaky-Gut Syndrome.” submitted to International Journal of Molecular Sciences. The reviewers’ comments and Editor’s comments were very encouraging and helpful. We have now amended our manuscript according to your suggestions and the Reviewer’s comments. Please find below the changes we have made to the manuscript and our answers to the reviewers’ comments in a point-by-point manner.
Response to Reviewer 2.
Comment 1.
Thanks to the addition of the molecular mechanism, I think it is easier to understand the pathogenesis of NAFLD. Not only SREBP-1c but also PPARgamma is involved in the pathogenesis of NAFLD. PPARalpha is also involved in preventing NAFLD development. Showing the relationship between these molecules in the liver and the small intestinal environment will make a more complete paper.
Response to comment 1.
I appreciate your advice. We added new chapter regarding about PPARs, NAFLD and small intestinal environment (Chapter 8).
